# Augmenting Vacuolar H^+^-ATPase Function Prevents Cardiomyocytes from Lipid-Overload Induced Dysfunction

**DOI:** 10.3390/ijms21041520

**Published:** 2020-02-23

**Authors:** Shujin Wang, Li-Yen Wong, Dietbert Neumann, Yilin Liu, Aomin Sun, Gudrun Antoons, Agnieszka Strzelecka, Jan F.C. Glatz, Miranda Nabben, Joost J.F.P. Luiken

**Affiliations:** 1Department of Genetics & Cell Biology, Faculty of Health, Medicine and Life Sciences, Maastricht University, 6200-MD Maastricht, The Netherlands; shujin.wang@maastrichtuniversity.nl (S.W.); l.wong@maastrichtuniversity.nl (L.-Y.W.); liuyilin8796@outlook.com (Y.L.); a.sun@maastrichtuniversity.nl (A.S.); a.strzelecka@maastrichtuniversity.nl (A.S.); m.nabben@maastrichtuniversity.nl (M.N.); 2Department of Clinical Genetics, Maastricht University Medical Center+, 6200-MD Maastricht, The Netherlands; 3Departments of Pathology, CARIM School for Cardiovascular Diseases, Maastricht University, 6200-MD Maastricht, The Netherlands; d.neumann@maastrichtuniversity.nl; 4Departments of Physiology, CARIM School for Cardiovascular Diseases, Maastricht University, 6200-MD Maastricht, The Netherlands; Gudrun.Antoons@UGent.be

**Keywords:** vacuolar H^+^-ATPase, lipid accumulation, insulin resistance, contractile function, diabetic heart

## Abstract

The diabetic heart is characterized by a shift in substrate utilization from glucose to lipids, which may ultimately lead to contractile dysfunction. This substrate shift is facilitated by increased translocation of lipid transporter CD36 (SR-B2) from endosomes to the sarcolemma resulting in increased lipid uptake. We previously showed that endosomal retention of CD36 is dependent on the proper functioning of vacuolar H^+^-ATPase (v-ATPase). Excess lipids trigger CD36 translocation through inhibition of v-ATPase function. Conversely, in yeast, glucose availability is known to enhance v-ATPase function, allowing us to hypothesize that glucose availability, via v-ATPase, may internalize CD36 and restore contractile function in lipid-overloaded cardiomyocytes. Increased glucose availability was achieved through (a) high glucose (25 mM) addition to the culture medium or (b) adenoviral overexpression of protein kinase-D1 (a kinase mediating GLUT4 translocation). In HL-1 cardiomyocytes, adult rat and human cardiomyocytes cultured under high-lipid conditions, each treatment stimulated v-ATPase re-assembly, endosomal acidification, endosomal CD36 retention and prevented myocellular lipid accumulation. Additionally, these treatments preserved insulin-stimulated GLUT4 translocation and glucose uptake as well as contractile force. The present findings reveal v-ATPase functions as a key regulator of cardiomyocyte substrate preference and as a novel potential treatment approach for the diabetic heart.

## 1. Introduction

The healthy myocardium maintains a balance between glucose and lipids as a nutrient source. In contrast, the diabetic heart is characterized by a shift in substrate utilization from glucose to lipids, resulting in insulin resistance and impaired cardiac contractile function [1,2]. Notably, at the level of cardiomyocytes, this is accompanied by increased abundance of CD36, the predominant cardiac lipid transporter, at the sarcolemma [3]. In the healthy heart, CD36 stimulates long chain fatty acid (LCFA) uptake via reversible translocation from the endosomes to the sarcolemma upon hormonal or mechanical stimuli [1]. Upon long-term overexposure of the heart to lipids, CD36 chronically translocates to the sarcolemma, initiating a vicious cycle of increased lipid uptake and lipid-induced insulin resistance, leading to cardiac dysfunction [4]. Our previous work showed that CD36 was expelled from the endosomes to translocate to the sarcolemma upon treatment of cardiomyocytes with pharmacological agents (e.g., monensin, and bafilomycin-A) causing endosomal alkalinization, indicating that proper functioning of the endosomal proton pump vacuolar H^+^-ATPase (v-ATPase) is needed for endosomal CD36 retention [5,6]. Studies in high-fat diet fed rats demonstrated that increased myocellular CD36 abundance at the sarcolemma of cardiomyocytes, as well as increased LCFA uptake and decreased insulin signaling was due to loss of endosomal acidification, thereby providing a novel link between v-ATPase inhibition and decreased cardiac function in the lipid-overloaded diabetic heart [6]. Taken together, v-ATPase inhibition underlies increased CD36 translocation to the sarcolemma and subsequent development of lipid-induced cardiomyopathy. 

As a proton pump that is present in acidic organelles, v-ATPase is responsible for endosomal acidification [7]. v-ATPase is structurally divided into a cytosolic V_1_ sub-complex and a transmembrane V_0_ sub-complex, encompassing the ATP catalyzing activity and the proton channel, respectively [8]. Studies in yeast and mammalian kidney cells revealed that v-ATPase activity is mainly regulated via assembly and disassembly of the two sub-complexes [9,10,11]. These assembly/disassembly cycles were found to be regulated by glucose availability, as glucose deprivation caused v-ATPase disassembly, whereas glucose-enriched conditions favored assembly and hence restoration of organellar acidification [9,10,11]. Upon disassembly of the V_1_ and V_0_ sub-complexes, such as when cells are subjected to glucose withdrawal, v-ATPase activity declines [12]. Similarly to glucose deprivation, lipid-overload induces v-ATPase disassembly in HL-1 cardiac cells and primary adult rat cardiomyocytes (aRCMs), mechanistically underlying CD36 translocation [6]. Together, these observations suggest that v-ATPase assembly/disassembly cycles are reciprocally regulated by glucose and lipids. 

Based on the experiments in yeast where increased glucose availability results in reassembly of v-ATPase [9,10,11], we hypothesize that reassembly of v-ATPase via forced glucose influx in lipid-overloaded cardiomyocytes will internalize CD36, decrease fatty acid uptake, improve insulin sensitivity, and restore contractile function. To test whether forced glucose uptake can preserve contractile function in lipid-overloaded cardiomyocytes, we applied two conditions: (i) High glucose concentration (25 mM) in the culture media, and (ii) adenoviral overexpression of protein kinase-D1 (PKD1). With respect to condition (i), the increased glucose concentrations will be taken up by GLUT1 and GLUT4, the two most highly expressed glucose transporters in cardiomyocytes, of which GLUT1 mediates basal glucose transport whereas GLUT4 is mainly responsible for insulin or contraction-induced glucose transport [13,14]. Recent studies showed a direct effect of 25 mM glucose exposure on GLUT1 expression, thus contributing to the increased glucose uptake [15,16]. Condition (ii) needs further explanation. PKD1, a member of a novel class of Ser/Thr kinases belonging to the PKD family, is specifically involved in regulation of GLUT4 translocation but not CD36 translocation [17]. Accordingly, cardiac-specific overexpression of constitutively active PKD1 in mice resulted in increased GLUT4 translocation and a shift in cardiac substrate preference from fatty acids to glucose [18]. 

The present data show that applying the two forced glucose uptake conditions to lipid-overloaded cardiomyocytes each resulted in increased v-ATPase assembly, proper endosomal acidification, endosomal CD36 retention, lowered levels of myocellular lipids and normalized contractile function. We further observed partial recovery of insulin-stimulated GLUT4 translocation in the absence of detectable effects on insulin signaling. Accordingly, targeting v-ATPase assembly as a possible strategy to combat lipid-induced cardiomyopathy warrants further investigation.

## 2. Results

### 2.1. Forced Glucose Uptake Re-Assembles v-ATPase in Lipid-Overloaded Cardiomyocytes

Our earlier studies showed that lipid-overexposure (or high-palmitate treatment) inhibits v-ATPase function via dissociation of the soluble V_1_ sub-complex from the endosomal membrane-bound V_0_ sub-complex, thereby providing a mechanistic explanation for lipid-induced endosomal alkalinization [6]. Using a subcellular fractionation method, we confirmed in aRCMs (Figure 1A,B) and HL-1 cells (Appendix A) that in all the conditions the V_0_-a2 subunit, as part of the membrane-bound V_0_ sub-complex, was localized at the membrane, whereas upon lipid-overexposure the V_1_-B2 subunit, as indicator of the soluble V_1_ sub-complex, was shifted from the endosomal membrane to the cytoplasmic fraction. Upon high glucose treatment, V_1_-B2 was redistributed back to the membrane fraction (aRCMs: Figure 1A,B; HL-1 cells: Appendix A). In a complementary approach, we studied the influence of forced glucose influx on v-ATPase V_0_/V_1_ assembly by applying glucose addition followed by IP of v-ATPase sub-complexes. Using antibodies against the d1 subunit of the V_0_ super-complex (V_0_-d1) or V_1_-B2 as part of the soluble V_1_ super-complex of v-ATPase (Figure 1C,D), we verified that lipid-overexposure conditions lead to a lower degree of co-IP with the other subunits as compared to control condition, which indicated increased V_0_/V_1_ disassembly upon lipid overload. Conversely, the degree of co-immunoprecipitation with the other subunits (e.g., V_0-_d1 or V_1_-B2) increased upon-treatment of lipid-overloaded cells with high glucose, indicating that forced glucose influx reassembled V_0_/V_1_ status in lipid-overloaded cardiomyocytes (Figure 1C,D). Together, these findings show that increased glucose influx promotes re-assembly of v-ATPase in lipid-overloaded cardiomyocytes.

### 2.2. Reassembly of V_0_/V_1_ Restores Endosomal Acidity in Lipid-overloaded Cardiomyocytes

To further investigate whether forced glucose influx (via high glucose or AdPKD overexpression) can restore proper endosomal acidification, we measured [^3^H]CHLQ accumulation as an indicator of v-ATPase activity. Pharmacological inhibition of v-ATPase by bafilomycin-A (BafA) caused >80% decrease of v-ATPase function in both aRCMs (Figure 1E,F) and HL-1 cells (Appendix A), consistent with our previous findings [6], while v-ATPase function was also reduced by >30% in lipid (high-palmitate)-overloaded aRCMs (Figure 1E,F) and >50% in lipid-overloaded HL-1 cells (Appendix A). When high-palmitate exposure was combined with high glucose treatment or AdPKD treatment, v-ATPase function was not significantly decreased compared to the “LP’ or “AdGFP’ condition.

### 2.3. Increased Endosomal Acidification Induces Endosomal CD36 Retention and Decreases Lipid Accumulation in Lipid-overloaded Cardiomyocytes

Our earlier results showed that v-ATPase inhibition leads to increased CD36 translocation from endosomes to the sarcolemma [6]. Here, we further assessed whether the effects of forced myocardial glucose influx on v-ATPase function could be further extended to the regulation of CD36 translocation. HP treatment of HL-1 cells lead to a ~2-fold upregulation in basal cell surface CD36 content, indicative of CD36 translocation (Figure 2A). A short-term stimulation by insulin in this condition of lipid overload had no additive effect on cell surface CD36 content indicating insulin resistance (Figure 2A and Appendix A). Treatment of palmitate-overexposed HL-1 cells with high glucose prevented the lipid-overload induced CD36 translocation to the sarcolemma and restored insulin-induced CD36 translocation in this condition (Figure 2A and Appendix A). 

Furthermore, we tested the effect of increased glucose influx on cellular lipid accumulation (e.g., triacylglycerol content). As expected, myocellular triacylglycerol content was increased (i.e., HL-1 cells; by 1.8-fold; aRCMs: by 3.2-fold) in lipid-overloaded cardiomyocytes (Figure 2B,C). High glucose treatment of these cells significantly prevented this increase. Collectively, these findings indicate that the beneficial effects of forced glucose uptake on v-ATPase function in lipid-overloaded cells extend to the endosomal retention of CD36, and to a decrease in triacylglycerol accumulation. 

### 2.4. Reassembly of v-ATPase Does Not Promote Insulin Sensitivity in Lipid-overloaded Cardiomyocytes

For evaluation of insulin signaling, phosphorylation levels of Akt (pAkt Ser473) and of ribosomal protein S6 (pS6 Ser235/236), were assessed. In agreement with previous observations [6], lipid-overload in aRCMs and HL-1 cells induced a loss of insulin-stimulated phosphorylation of Akt and S6. However, both glucose uptake-enforcing treatments (aRCMs: Figure 3A–F; HL-1 cells: Appendix A) did not restore this lipid-induced loss of insulin-stimulated Akt phosphorylation and S6 phosphorylation.

### 2.5. Reassembly of v-ATPase Improves Insulin Stimulated-GLUT4 Translocation and Glucose Uptake in Lipid-overloaded Cardiomyocytes

A surface biotinylation assay was used to investigate whether insulin-stimulated GLUT4 translocation would be improved in lipid-overloaded cardiomyocytes upon treatment with either high glucose exposure or AdPKD overexpression. For obtaining information about GLUT4 translocation, we assessed the cell surface content of insulin-responsive aminopeptidase (IRAP). IRAP is an abundant cargo protein associated with GLUT4 vesicles that translocates in response to insulin in a manner identical to GLUT4 [19]. Cell surface content of IRAP was largely decreased upon HP treatment and almost entirely re-installed in lipid-overloaded aRCMs under glucose addition (Figure 4A,B, and Appendix A) and AdPKD overexpression (Figure 4C,D and Appendix A). 

A radiolabeled glucose analog ([^3^H]deoxyglucose) was used to investigate whether both treatments could preserve insulin-stimulated glucose uptake in lipid-overloaded cardiomyocytes. PKD overexpression caused an increase in glucose uptake (1.8 fold in aRCMs and 1.4-fold in HL-1 cells), which is due to increased insulin-independent GLUT4 translocation [18]. As expected, insulin-stimulated glucose uptake was almost entirely abolished in high palmitate-treated cells (aRCMs: Figure 4E,F and Appendix A; HL-1 cells: Appendix A). In both cell models, this loss of insulin-stimulated glucose uptake was partly corrected by each of the treatments (aRCMs: Figure 4E,F and Appendix A; HL-1 cells: Appendix A). Taken together, at the level of insulin-stimulated GLUT4 translocation as well as at the level of insulin-stimulated glucose uptake, both treatments proved to be at least partly effective in restoring the relatively large insulin effect of the low-palmitate condition. 

### 2.6. Reassembly of v-ATPase Restores Contractile Function in Lipid-overloaded Cardiomyocytes

Given our previous finding that high lipid exposure causes contractile dysfunction, we next tested if high glucose exposure could restore such loss. Indeed, as seen previously [6], lipid induced-v-ATPase inhibition significantly decreased sarcomere shortening (Figure 5A), but it had no significant influence on contraction acceleration time (e.g., time to peak) and the duration of relaxation (e.g., decay time to 50% percent (RT50), and decay time to 90% percent (RT90)) in lipid-overloaded aRCMs (Figure 5B–D). Notably, treatment of these lipid-overloaded aRCMs with high glucose could restore a decrease of sarcomere shortening (Figure 5A), also without changes in both the contraction acceleration time (e.g., time to peak) (Figure 5B) and the duration of relaxation (e.g., RT50 and RT90) (Figure 5C,D). Collectively, treatment of these lipid-overloaded aRCMs with high glucose could restore their contractile function.

### 2.7. Restoration of v-ATPase Function Re-balances Energy Substrates in Human iPSC-Cardiomyocytes

To further confirm whether the rebalancing of cellular energy substrate metabolism via restoration of v-ATPase function also occurs in human iPSC-cardiomyocytes, hiPSC-CMs were subsequently used for this study. These hiPSCs were first characterized for their pluripotency prior to cardiomyocyte differentiation and as shown in Appendix A, the pluripotent markers (e.g., Nanog and Oct4) were highly expressed in hiPSCs. Additionally, the karyotyping results of hiPSCs from the Department of Clinical Genetics, Maastricht UMC+, also demonstrated that there were no chromosomal abnormalities observed in these cells (Appendix A). 

Similar to that observed in rodent cardiomyocytes (aRCMs: Figure 4E,F and Appendix A; HL-1 cells: Appendix A), hiPSC-CMs developed the key features of insulin resistance upon high-palmitate culturing: loss of insulin-stimulated LCFA and glucose uptake and increased basal LCFA uptake (Figure 6B,C and Appendix A). Moreover, hiPSCs-CM cultured with high-palmitate display loss of v-ATPase function (Figure 6A). When high-palmitate exposure was combined with high glucose treatment, v-ATPase function, insulin-stimulated LCFA- and glucose uptake were restored (Figure 6A–C). Most importantly, these findings strongly support the observations seen in rodent cardiomyocytes and provide substantial evidence that the mechanism of increased glucose availability-driven v-ATPase assembly to re-balance substrate uptake in lipid-overloaded cardiomyocytes is conserved between species.

## 3. Discussion

In this study, we investigated whether forced glucose uptake would improve insulin resistance and contractile dysfunction in lipid-overloaded cardiomyocytes and whether this is accompanied by re-assembly of v-ATPase. The following main observations were made (Figure 7): High glucose addition and AdPKD overexpression stimulated assembly of v-ATPase, endosomal acidification, endosomal CD36 retention, and insulin-stimulated glucose uptake, inhibited lipid accumulation, and promoted contractile force in lipid-overloaded cardiomyocytes, but did not restore insulin signaling. 

As established earlier, lipid overload in cardiomyocytes leads to disassembly of the V_0_/V_1_ holo-complex [6]. The consequent inhibition of v-ATPase function causes CD36 translocation to the sarcolemma, leading to the well-described shift of substrate uptake from glucose to lipids, as seen in the diabetic heart [1,6]. Given that v-ATPase responds to the availability of lipids by dissociation of the super-complexes thereby decreasing its pumping function, v-ATPase could be seen as a lipid sensor. Vice versa, it has been shown in yeast that glucose can increase v-ATPase function by restoring the assembly of its super-complexes [8,9]. Here, we set out to investigate the effect of forced glucose uptake in lipid-overloaded mammalian cardiomyocytes, which is a setting where glucose and lipids would compete for v-ATPase assembly and activity. Indeed, the present study shows that lipid-induced dissociation of v-ATPase was partially prevented by forced glucose uptake. Consistent with this, forced glucose uptake also prevented the loss of v-ATPase activity, decreased CD36 translocation and cellular lipid accumulation, and reinstalled insulin-stimulated glucose uptake as well as contractile function. Reassembly of v-ATPase is therefore deemed as a suitable target to counteract lipid accumulation in the diabetic heart. 

High glucose treatment by itself, especially in a diabetic setting, would be a truly counter-intuitive strategy to combat lipid-induced insulin resistance. Therefore, we would like to stress that glucose addition was chosen to provide proof-of-principle for testing strategies to counteract lipid-induced contractile dysfunction via v-ATPase re-assembly. The other strategy to enforce glucose uptake, i.e., PKD1 overexpression, would be more favorable in the diabetic setting because it stimulates glucose entry into cardiomyocytes via induction of contraction-responsive GLUT4 not CD36 translocation, which would even contribute to lowering the circulating glucose levels [18]. This favorable anti-diabetic action would be even greater when PKD1 would induce GLUT4 translocation also in skeletal muscle, but the latter is currently undefined. Hence, as a future step in therapeutic strategies against lipid-induced insulin resistance, specific PKD1 activators could be employed to coerce myocytes to specifically take up glucose for subsequent v-ATPase reassembly. A number of compounds, such as α-adrenergic agonists, are known to induce PKD1 activation [20], but also to induce a variety of other signaling pathways with rather undesirable side actions such as cardiac hypertrophy. Although specific PKD1 activators have not been identified to date, its development could hold promise for future therapy in diabetic heart. Without question further studies are needed to advance the findings of the applied short-term v-ATPase activation towards a possible treatment regime of a chronic disease state such as diabetes, which should include suitable in vivo approaches using (pre-)diabetic models. 

Surprisingly, we did not observe positive effects of forced glucose uptake on insulin-stimulated Akt phosphorylation. This raises the question on how glucose uptake-enforcing treatments can restore insulin-stimulated glucose uptake in the lipid-overloaded heart in the absence of restoration of insulin signaling. A possible explanation includes that lipids negatively impact on insulin-stimulated glucose uptake in the heart independently of impairment of insulin-stimulated Akt activation. In this respect, an alternative mechanism of lipid-induced impairment of insulin-stimulated glucose uptake involves the competition between GLUT4 vesicles and lipid droplets for the same SNARE proteins. Recently, it has been revealed that lipid droplets use SNARE proteins for dynamic fission/fusion processes [21]. One of the SNARE proteins shared by GLUT4 vesicles and lipid droplets is SNAP23 [21]. Using this information, the following scenario could take place when the lipid-overloaded cells are forced to take up glucose. First, v-ATPase would re-assemble leading to endosomal re-acidification, CD36 endocytosis, and decreased myocellular lipid uptake. The consequential decline in lipid droplets would then make SNAP23 re-available for insulin-stimulated GLUT4 translocation without the need for stimulation through Akt signaling.

Yet there would still be one unresolved issue with this SNARE competition-driven scenario in lipid-overloaded cells: There is still a ~50% decrease in insulin-stimulated Akt phosphorylation, which is upstream of insulin-stimulated GLUT4 translocation (via phosphorylation and inhibition of AS160), and as mentioned, not restored upon PKD1 overexpression. This issue can be circumvented when it would be assumed that the partial impairment of Akt activation would not be rate limiting for insulin-stimulated GLUT4 translocation, as has been suggested by us earlier [4].

## 4. Materials and Methods 

### 4.1. Antibodies

Primary antibodies used in Western blotting analysis were rabbit anti- PKD/PKC-μ, anti-total AKT (T-AKT), rabbit anti-phospho-Ser473-AKT (pAKT), rabbit anti- phospho-Ser235/236-S6 (pS6), rabbit anti- IRAP, and rabbit anti-GAPDH (Cell Signaling Technology, Danvers, MA, USA), rabbit anti-ATP6V_0_A2 (V_0_-a2), rabbit anti-ATP6V_0_D1 (V_0_-d1), rabbit anti-ATP6V_1_B2 (V_1_-B2) (Abcam, Cambridge, UK), mouse anti-CD36 (MO25) (a generous gift from Dr. N. Tandon), mouse anti- caveolin-3 (BD Transduction Laboratories, Lexington, KY, USA). Primary antibodies were detected by either anti-rabbit secondary antibody for PKD/PKC-μ, T-AKT, pAKT, pS6, IRAP and GAPDH (Cell Signaling Technology) or anti-rabbit secondary antibody for V_0_-a2, V_0_-d1, and V_1_-B2 (Dako Corp., Carpinteria, CA, USA), or anti-mouse secondary antibody for caveolin-3 (BD Transduction Laboratories Dako Corp., Carpinteria, CA, USA).

### 4.2. Isolation and Culturing of Primary Rat Cardiomyocytes 

Male Lewis rats, 250–340 g, were purchased from Charles River laboratories, and were maintained at the Experimental Animal Facility of Maastricht University. Animals were housed in a controlled environment (21–22 °C) on a 12:12 h light dark cycle (light from 0700 to 1900 h) and had free-access to food and tap water. All animal experiments (UM-project license number: PV-2016-004) were performed according to Dutch regulations and approved by the Maastricht University Committee for Animal Welfare. 

Adult rat cardiomyocytes (aRCMs) were isolated by using a Langendorff perfusion system, as previously described [22]. Briefly, after isolation of cardiomyocytes, cells were seeded on laminin pre-coated plates. After a 2 h adhesion period the medium was replaced by either low palmitate medium (LP, 20 μM palmitate, palmitate/BSA ratios of 0.3:1), LP with the addition of 25 mM glucose (LP/HG), high palmitate medium (HP, 200 μM palmitate, palmitate/BSA ratios of 3:1), or HP with the addition of 25 mM glucose (HP/HG) for 24 h. Cells were cultured as previously described [6].

### 4.3. Culturing of HL-1 Cardiomyocytes

HL-1 cells were kindly provided by Dr. W. Claycomb (Louisiana State University, New Orleans, LA, USA) and cultured as previously described [6]. Briefly, HL-1 cells were either treated with control (Ctrl) medium, Ctrl medium containing 100 nM Bafilomycin-A (BafA), HP medium containing 500 µM palmitate and 100 nM insulin (HP), or HP medium with 25mM glucose addition (HP/ HG) for 24 h. 

### 4.4. Human Induced Pluripotent Stem Cell (hiPSC) Maintenance and Differentiation into Cardiomyocytes (hiPSC-CMs)

Skin fibroblasts from healthy adult male individuals were collected and hiPSCs were generated by episomal reprogramming at the Stem Cell Technology Centre, Radboudumc (Nijmegen, Netherlands). The cells were maintained in Essential 8 medium (Thermofisher Scientific, Miami, Florida, USA) under feeder-free conditions. Prior to cardiomyocyte differentiation, the cells were passaged with 0.5 mM of EDTA solution (Promega, Madison, Wisconsin, USA), counted with a cell-counter and seeded in Essential 8 medium containing 10 µM ROCK inhibitor (Stem Cell Technologies, Vancouver, Canada) on Matrigel (Corning Inc., NY, USA)-coated 24-well and 12-well plates. This was denoted as day 4 and the medium was changed daily until the cells reached 80%–90% confluency. At day 1 of differentiation, Essential 8 medium was removed and replaced with Cardiomyocyte Medium A (Thermofisher Scientific, Miami, Florida, USA). The cells were incubated for 2 days prior to a media change to Cardiomyocyte Medium B (Thermofisher Scientific, Miami, Florida, USA). At day 5, the media was then replaced with Cardiomyocyte Maintenance Media (Thermofisher Scientific, Miami, Florida, USA) for 2 days and replaced every 2 days until day 14. To purify the cardiomyocyte population, metabolic selection was performed using RPMI 1640 without glucose (Thermofisher Scientific, Miami, Florida, USA) containing 4 mM sodium lactate (Sigma Aldrich, St Louis, Missouri, USA) for 5 days and thereafter the cells were further maintained in Cardiomyocyte Maintenance Media for an additional 5 days. Chloroquine accumulation, fatty acid and glucose uptake assays were subsequently performed on the metabolically-selected hiPSC-CMs.

### 4.5. Adenovirus Amplification

Adenovirus Containing Green Fluorescent Protein (AdGFP, as control) and a recombinant adenovirus encoding full-length wild-type mouse PKD1 (AdPKD) were kindly provided by Dr. L. Bertrand’s lab (Pole de Recherche Cardiovasculaire, Institut de Recherche Expérimentale et Clinique, UCLouvain, Brussels, Belgium). Briefly, adenoviruses were amplified in HEK-293 cells and then purified over CsCl2 gradients as previously described [23]. Optimal multiplicity of infection (MOI) was determined by fluorescence microscopy, and MOI 10 was therefore chosen for AdPKD [23].

For the transfection in aRCMs, cells were transfected with either AdGFP or AdPKD for 48 h after a 2 h adhesion period. Additionally, for the transfection in HL-1 cells, they were also transfected with AdGFP or AdPKD. After 32 h, the medium was changed into FCS- and norepinephrine-free medium and cells were kept overnight. Subsequently, cells were used for measurements of insulin signaling, substrate uptake, or cellular chloroquine accumulation.

### 4.6. Measurement of v-ATPase Disassembly/Assembly 

Two methods were applied to measure disassembly, namely, immunoprecipitation (IP) (i) and subcellular fractionation (ii). 

(i) IP: The method of IP was conducted as previously described [6]. v-ATPase d1 (V_0_-d1, an indicator of cytosolic V_1_ sub-complex) and v-ATPase B2 (V_1_-B2, an indicator of the membrane-bound V_0_ sub-complex) proteins were detected by Western blotting.

(ii) Subcellular fractionation: the method of subcellular fractionation was also conducted as previously described [6]. For subcellular fractionation, V_0_-a2, V_0_-d1, and V_1_-B2, GAPDH, and caveolin-3 proteins were detected by Western blotting. 

Measurement of Cellular Chloroquine (CHLQ) Accumulation As Readout of v-ATPase Function. 

CHLQ accumulation in HL-1 cells, aRCMs, and hiPSC-CMs was measured as previously described [5,6].

### 4.7. Determination of Content of CD36 at Cell Surface

Cells were incubated with an anti-CD36 antibody, washed and subsequently incubated with a horseradish peroxidase (HRP)-linked secondary antibody. After further washing steps, ortho-phenylenediamine-H_2_O_2_ was added as a substrate for the bound HRP. Upon termination of the reaction, color development (representing CD36 content at the cell surface) was quantified by measurement of the absorbance at 490 nm. For more detail, see ref [24]. 

### 4.8. Quantification of Triacylglycerol 

Quantification of triacylglycerol in both HL-1 cells and aRCMs was performed using a Triglyceride Assay Kit (ab65336, Abcam, San Francisco, CA, USA) following the manufacturer’s instructions.

### 4.9. Determination of Insulin Sensitivity

After culturing, aRCMs and HL-1 cells were exposed to insulin (aRCMs, 100nm; HL-1 cells, 200 nM) for 30 min to be able to compare basal-phosphorylation to insulin-stimulated phosphorylation. Afterwards, these cells were lysed in sample buffer and used for protein detection by SDS-polyacrylamide gel electrophoresis, followed by Western blotting. PKD/PKC-μ, phospho-Ser473-Akt (pAKT), phospho-Ser235/236-S6 (pS6), and caveolin-3 proteins were detected by Western blotting.

### 4.10. Surface Biotinylation Assay

Surface-protein biotinylation was measured as previously described [25] with the modifications described below. After culturing with high palmitate medium or the transfection with AdPKD, aRCMs were incubated for 30 min with (or without) 100 nm insulin. Subsequently the cells were biotinylated with the cell-impermeable reagent sulfo-NHS-LC-biotin (0.5 mg/mL dissolved in M199 medium, Thermo Fisher Scientific, Fremont, CA, USA) for 5 min at 37 °C. The rest steps of surface-protein biotinylation assay were carried out as previously described [25]. Insulin-regulated aminopeptidase protein (IRAP, which reflects GLUT4 trafficking) was detected by Western blotting. 

### 4.11. Measurement of Substrate Uptake

[^3^H]Deoxyglucose uptake into suspensions of cardiomyocytes was measured as previously described [26]. Uptake of substrate into HL-1 cells and into cultured aRCMs and hiPSC-CMs, all cell models seeded on pre-coated glass slides, was measured as previously described [27].

### 4.12. Measurement of Cardiomyocytic Contraction Dynamics

Contractile properties of aRCMs were assessed at 1 Hz field stimulation using a video-based cell geometry system to measure sarcomere dynamics (IonOptix, Milton, MA, USA). From the digitized recordings acquired with IonWizard acquisition software, the following parameters were calculated: Sarcomere shortening, time to peak, and decay time. This was conducted as previously described [4].

### 4.13. Statistical Analysis 

All data are presented as means ± SEM. Statistical analysis was performed by using a two-sided Student’s t-test with GraphPad Prism 5 software (GraphPad Software, Inc., San Diego, CA, USA). *P*-values of less than 0.05 are considered statistically significant.

## 5. Conclusions

In summary, the overall findings provide solid evidence that v-ATPase is both a lipid sensor and a glucose sensor in the heart, making v-ATPase a key regulator of cardiac substrate preference. The present findings as observed in rodents also extend to the human setting, given that the enforcement of glucose uptake also preserves v-ATPase activity and insulin-stimulated glucose uptake treatment in hiPSC-CMs upon over-exposure of these cells to lipids. Hence, the regulation of v-ATPase assembly/disassembly may offer a suitable target to combat cardiac dysfunction elicited by lipotoxic conditions.

## Figures and Tables

**Figure 1 ijms-21-01520-f001:**
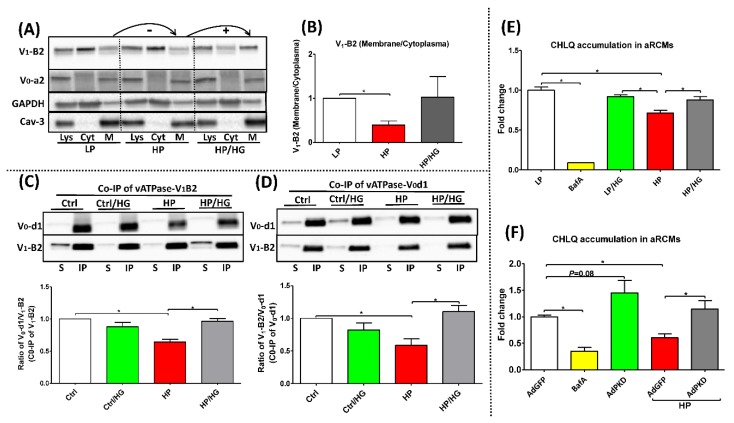
Status and activity of v-ATPase in lipid-overexposed cardiomyocytes. (**A**,**B**) Fractionation in adult rat cardiomyocytes (aRCMs): aRCMs were incubated for 24h with either low palmitate (LP, palmitate/BSA ratio 0.3:1), high palmitate (HP, palmitate/BSA ratio 3:1), or HP with the addition of 25mM glucose (HP/HG). Contents of v-ATPase subunit a2 (V_0_-a2) and subunit B2 (V_1_-B2) were assessed by Western blotting in total cell lysates (Lys), in the cytoplasmic fraction (Cyt) and in the membrane fraction (M). GAPDH and Caveolin-3 were detected as loading control for V_1_-B2 and V_0_-a2, respectively. (**A**) Representative blots of three independent experiments are shown. (**B**) Quantification: The ratio of V_1_-B2 (membrane/cytoplasm) (*n* = 3). (**C**,**D**) Immunoprecipitation (IP) of v-ATPase subunit d1 (V_0_-d1) or subunit B2 from HL-1 cells after incubation for 24 h with either basal (Ctrl) medium, Ctrl medium supplemented with 25 mM glucose (Ctrl/HG), HP medium containing 500µM palmitate and 100 nM insulin, or HP medium supplemented with 25 mM glucose (HP/ HG). IP samples were immunoblotted with antibodies against v-ATPase subunits V_0_-d1 and V_1_-B2 (*n* = 4). (**E**,**F**) Chloroquine (CHLQ) accumulation in lipid-overexposed aRCMs: (**E**) aRCMs were incubated for 24h with LP medium (LP), LP supplemented with 100nM Bafilomycin-A (BafA), LP/HG, HP, and HP/HG; (**F**) aRCMs were incubated for 48 h with either LP medium with the addition of 120 µL AdGFP (AdGFP), AdGFP supplemented with 100 nM Bafilomycin-A (BafA), LP medium with the addition of 120 µL AdPKD (AdPKD), HP with the addition of 120 µL AdGFP (AdGFP/HP), or HP with the addition of 120 µL AdPKD (AdPKD/HP). After the culturing of all conditions above, cells were sub for [^3^H] CQ accumulation assay last 20 min. Values are displayed as mean ± SEM (*n* = 4). * *p* < 0.05 were considered statistically significant.

**Figure 2 ijms-21-01520-f002:**
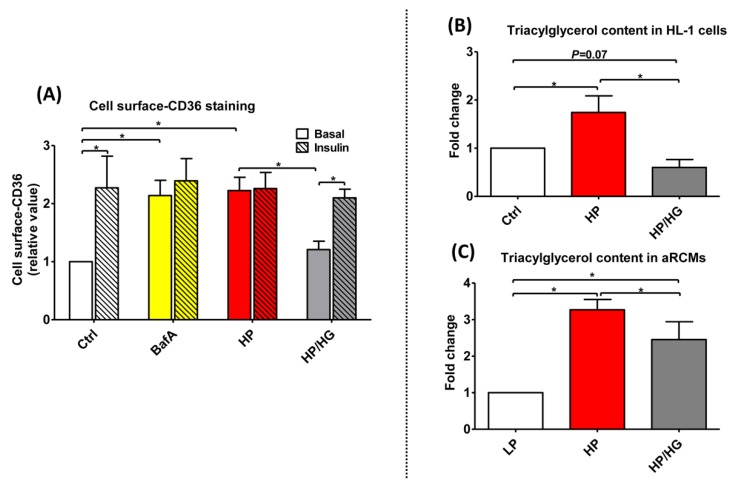
Cell surface-CD36 staining and triacylglycerol contents in lipid-overexposed cardiomyocytes. (**A**) Cell surface CD36 staining of HL-1 cells: Prior to CD36 cell surface staining, HL-1 cells were treated for 24 h either with control (Ctrl) medium, Ctrl medium containing 100 nM Bafilomycin-A (BafA), high palmitate medium containing 500 µM palmitate and 100 nM insulin (HP), or HP medium with 25 mM glucose (HP/HG). Subsequently, cells were stimulated either without or with 200 nM insulin for 30 min and immunochemically stained for cell surface CD36 content (*n* = 3). **(B-C)** Triacylglycerol contents in lipid-overexposed cardiomyocytes: (**B**) HL-1 cells were incubated for 24 h with either Ctrl medium, HP medium, or HP/HG (*n* = 5); (**C**) aRCMs were incubated for 24 h with either LP medium, HP medium, or HP/ HG (*n* = 7). Values are displayed as mean ± SEM. * *p* < 0.05 were considered statistically significant.

**Figure 3 ijms-21-01520-f003:**
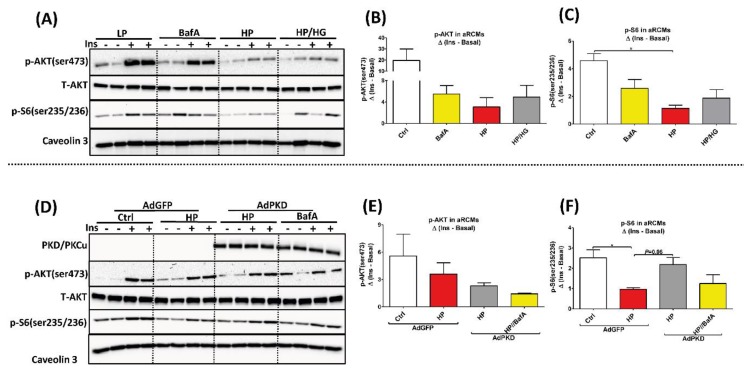
Insulin sensitivity in lipid overloaded aRCMs. (**A**–**C**) aRCMs were incubated for 24 h with either low palmitate (LP), LP medium containing 100nM Bafilomycin-A (BafA), high palmitate (HP), or HP with addition of 25 mM glucose (HP/HG) for 24 h. Subsequently, cells were stimulated either with or without 100 nM insulin for 30 min and harvested for Western blotting analysis. (**A**) Representative blots of pAKT, total-AKT, p-S6, and Caveolin 3. (**B**,**C**) Quantification of the level of pAKT and pS6 (*n* = 3). (**D**–**F**) aRCMs were incubated for 24 h in LP medium with either the addition of 120 µL AdGFP (AdGFP), HP with the addition of 120 µL AdGFP (AdGFP/HP), LP with the addition of 120 µL AdPKD (AdPKD), HP with the addition of 120 µL AdPKD (AdPKD/HP), or AdPKD/HP medium containing 100nM Bafilomycin-A (BafA) (AdPKD/HP+BafA). Subsequently, cells were stimulated either with or without 100 nM insulin for 30 min and harvested for Western blotting analysis. (**D**) Representative blots of PKD/PKCu, pAKT, total-AKT, p-S6, and Caveolin 3. (**E**,**F**) Quantification of the level of pAKT and pS6 (*n* = 3). Caveolin 3 was detected as loading control. Values are displayed as mean ± SEM. * *p* < 0.05 were considered statistically significant.

**Figure 4 ijms-21-01520-f004:**
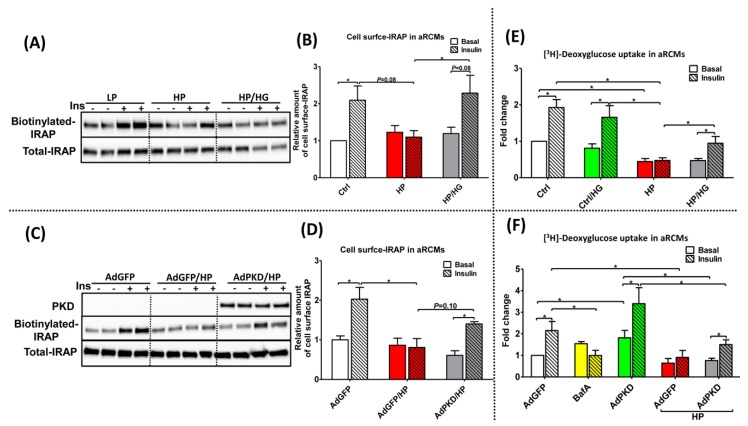
Insulin-stimulated GLUT4 translocation and glucose uptake in lipid-overexposed aRCMs. (**A**–**D**) Insulin-stimulated GLUT4 translocation in lipid-overexposed aRCMs: (**A**) Representative blotting and (**B**) quantification of Insulin-regulated aminopeptidase (IRAP, which reflects GLUT4 trafficking) from biotin-labeled (A upper panel) and total lysate fractions (A lower panel) of aRCMs, which are incubated for 24 h with low palmitate (LP), high palmitate (HP), and HP+25 mM glucose (HP/HG), followed by 30 min of (−/+) insulin (100 nM) incubation prior to biotin labeling and lysis (*n* = 4). (**C**) Representative blotting and (**D**) quantification of protein kinase-D1 (PKD) (A upper panel) and IRAP from biotin-labeled (A middle panel) and total lysate fractions (A lower panel) of aRCMs, which are incubated for 48 h with LP containing 120 µL Adenovirus Green Fluorescent Protein (AdGFP), HP containing 120 µL AdGFP (AdGFP/HP), and HP containing 120ul Adenovirus PKD (AdPKD/HP), followed by 30 min of (−/+) insulin (100 nM) incubation prior to biotin labeling and lysis (*n* = 4). Quantification of the amount of cell surface IRAP using BioRad Quantity One software. (**E**,**F**) Insulin-stimulated glucose uptake in lipid-overexposed aRCMs: (**E**) aRCMs were incubated for 24h with either LP, LP/HG, HP, or HP/HG, followed by 30 min of (−/+) insulin (100 nM) incubation prior to [^3^H] deoxyglucose labeling (*n* = 6); (**F**) aRCMs were incubated for 48h with AdGFP, AdGFP/HP, AdPKD, AdPKD/HP, or AdPKD/HP medium containing 100nM Bafilomycin-A (AdPKD/HP+BafA), followed by 30 min of (−/+) insulin (100 nM) incubation prior to [^3^H] deoxyglucose labeling (*n* = 4). Values are displayed as mean ± SEM. * *p* < 0.05 were considered statistically significant.

**Figure 5 ijms-21-01520-f005:**
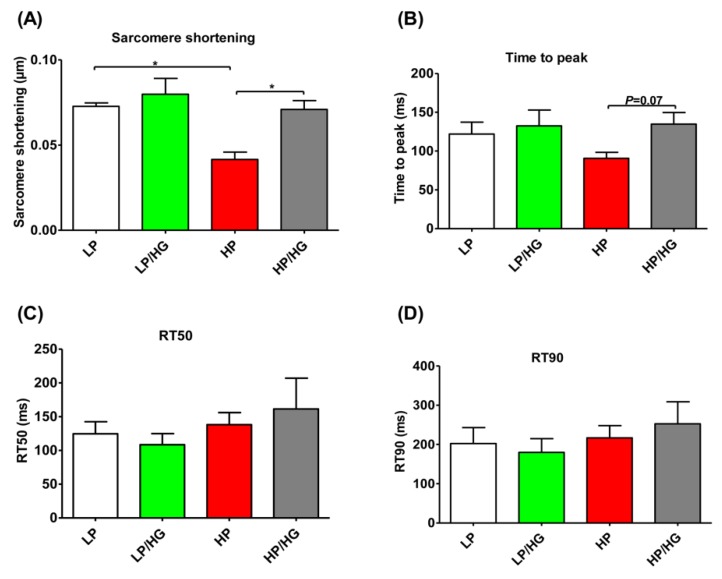
Contractile function in lipid overloaded aRCMs. (**A–D**) aRCMs were incubated for 24 h in either low palmitate medium (LP), LP medium with the addition of 25 mM glucose (LP/HG), high palmitate medium (HP), or HP with the addition of 25 mM glucose (HP/HG). Parameters of contraction amplitude and kinetics were determined upon 1 Hz electrostimulation: (**A**) sarcomere shortening; (**B**) time to peak; (**C**) decay time to 50% percent (RT50); (**D**) decay time to 90% percent (RT90). Values are displayed as mean ± SEM (*n* = 3; imaging of 10 cells/condition). * *p* < 0.05 were considered statistically significant.

**Figure 6 ijms-21-01520-f006:**
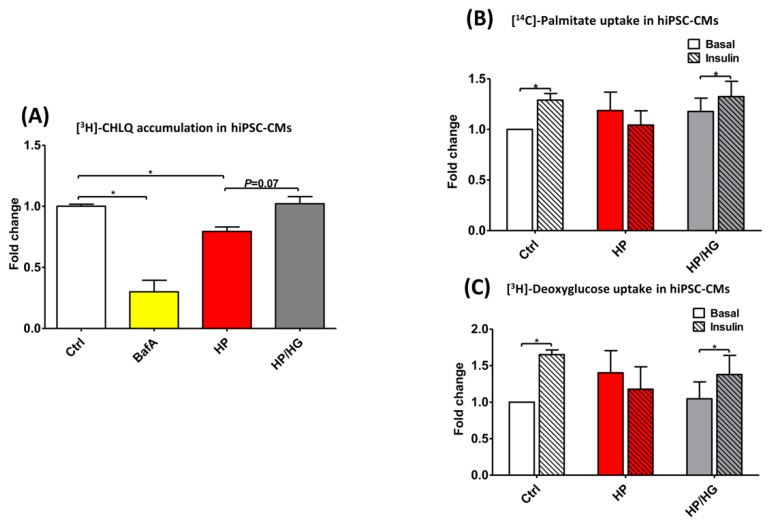
Chloroquine (CHLQ) accumulation, glucose- and fatty acid uptake in lipid-overexposed human iPSC-derived cardiomyocytes (hiPSC-CMs). (**A**) CHLQ accumulation in hiPSC-CMs cultured for 20 h in either control medium (Ctrl), Ctrl with addition of 100 nM Bafilomycin-A (BafA), high palmitate medium (HP), or HP with the addition of 25mM glucose (HP/HG). (**B**,**C**) Palmitate- and glucose uptake in in hiPSC-CMs treated without/with 100 nM insulin for 30 min, determined by means of [^14^C]palmitate and [^3^H]deoxyglucose uptake, respectively. Values are displayed as mean ± SEM (*n* = 3). * *p* < 0.05 were considered statistically significant.

**Figure 7 ijms-21-01520-f007:**
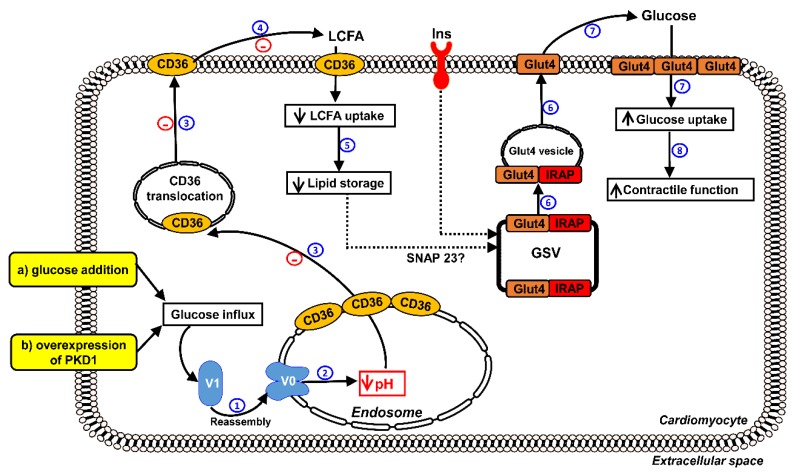
Schematic hypothesis of reassembly of v-ATPase as a target to restore contractile function in lipid-overloaded cardiomyocytes. When LCFA supply is high, CD36 translocation from the endosome to the sarcolemma is stimulated. Furthermore, the v-ATPase V_0_ sub-complex, which is integral to the endosomal membrane, is disassembled from the cytosolic V_1_ sub-complex contributing to endosomal alkalinization. In this situation, the supply of LCFA will be in excess to the immediate energy demand, resulting in insulin resistance and impaired cardiac contractile function. Therefore, proper endosomal acidification mediated by v-ATPase is essential for CD36 retention. Elevated glucose influx triggers a series of following events: (**1**) both treatments (e.g., glucose addition and overexpression of PKD1) promote the V_1_ and V_0_ sub-complexes to reassemble, V_1_ is therefore integrated to the membrane. (**2**) Reassembly of v-ATPase leads to endosomal acidification. (**3**) Endosomal acidification triggers CD36 retention to the endosome as well as Glut4 translocation to the sarcolemma. (**4**,**5**) The inhibition of CD36 translocation decreases LCFA uptake, thereby inhibiting lipid accumulation. (**6**,**7**) Decreased lipid droplet numbers leads to redistribution of SNAP23 to the GLUT4 storage vesicle (GSV) to become available for insulin-stimulated GLUT4 translocation and insulin-stimulated glucose uptake, therefore shifting energy substrate preference from LCFA to glucose. (**8**) Rebalancing of energy substrate utilization restores contractile function in lipid-overloaded cardiomyocytes.

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
