# Peer review of "Augmenting Vacuolar H+-ATPase Function Prevents Cardiomyocytes from Lipid-Overload Induced Dysfunction"

_ijms, 2020, doi:10.3390/ijms21041520_

Round 1

Reviewer 1 Report

General
In this review, authors declare vacuolar H+-ATPase function prevents cardiomyocytes from lipid-overload induced dysfunction. Cardiomyocytes cultured under high-lipid conditions, each treatment stimulated v-ATPase re-assembly, endosomal acidification, endosomal CD36 retention and prevented myocellular lipid accumulation. They addressed that these treatments preserved insulin-stimulated GLUT4 translocation and glucose uptake. The available information gives an interesting study and suggests possible mechanisms for glucose uptake and subsequent intracellular trafficking of proteins. I have formulated below a few points that could help to improve the clarity of some results.

Specific points

In figure 2a, low data images would be helpful for convincing the data. In figure 4, even if IRAP reflects GLUT4 trafficking, there are several methods to show the GLUT4 surface expression or trafficking such as biotinylation or confocal microscope with using GLUT4 antibody. Please use and show the GLUT expression to support your data. Please show the summary bar graph of Figure 1a

Typos: line 118, ardiomyocytes -> cardiomyocytes

Reviewer 2 Report

The manuscript by Wang and colleagues tests the hypothesis that glucose availability, via a mechanism utilizing the dynamics of vacuolar H+-ATPase (v-ATPase), internalizes the lipid transporter CD36, which in turn restores contractile function to cardiomyocytes in the context of lipid-overload. The study specifically tested this in primary adult rat ventricular cardiomyocytes (aRCM), immortalized mouse atrial cardiomyocytes (HL-1), or human induced pluripotent stem cell derived cardiomyocytes (hiPSC). Depending on the cell type, cultures were treated with various high glucose, high palmitate, and combinations of the two. To specifically stimulate glucose uptake some cells were further stimulated with overexpression of protein kinase-D1 (PKD1), which can stimulate glucose transporter 4 (GLUT4) translocation. Taken together the characterization of molecular changes, changes in cellular physiology, and improvement in contractile function support that v-ATPase works as both a cellular lipid and glucose sensor to mediate CD36. This results in altered lipid uptake which in turn regulates glucose utilization. Therefore, the authors conclude that regulation of v-ATPase may represent a molecular target for the treatment of cardiomyocyte dysfunction in the context of lipotoxicity. Taken together the results of the current study, combined with prior literature by this group, as well as studies focused on v-ATPase assembly/disassembly cycles in other organisms and cell-types, supports that the reciprocal regulation of glucose and lipids in cardiomyocyte regulation may impart be v-ATPase/CD36 driven. Overall, this is a well-written manuscript focused on the important areas of cardiac metabolism and its dysregulation in the disease state. As it pertains to novelty of the current study, the stark contrast of no change in Akt phosphorylation in Fig. 3 versus the restored IRAP and 2DG uptake in Fig. 4 with the HP/HG as well as AdPKD suggests a unique parallel pathway in support of the paper’s conclusions. Despite the somewhat more modest effects in the hiPSC, the overall conclusions of the authors are consistent. Only one minor point could be addressed to further improve this manuscript.

Comment: The authors point out in numerous places that these are short term studies. Although the data is convincing for this mechanism in cell culture in this timeframe the discussion should be tempered for relating to treatment in diabetes, a long-term chronic state. It remains possible based on the literature surrounding gluco/lipo-toxicity, which more accurately reflects the diabetic state, that secondary mechanisms would not be overcome by short-term treatment nor would increasing glucose in the context of chronic hyperglycemia be as robust in restoring function. Indeed, some studies specifically highlight that although lipotoxicity is present in diabetes adding glucose on top of that in the chronic state would actually exacerbate and not attenuate glucose uptake and signaling. This includes work by Rong Tian with the GLUT1 overexpressor, that in chronic nutrient overload actually leads to dysfunction as well as work of others.

Reviewer 3 Report

This study is well-done and its finding is interesting to understand pathology of diabetic heart. My question is how important the finding of your cardiomyocytic contraction dynamics experiment is in vivo. In my knowledge, diastolic dysfunction is more important in diabetic myopathy rather than systolic dysfunction. Do you have any information from in-vivo experiments to support the result of your in-vitro experiment ? 
